# Does a rare mutation in PTPRA contribute to the development of Parkinson's disease in an Australian multi-incident family?

**Melissa A. Hill[1], Steven R. Bentley[1], Tara L. Walker[2], George D. Mellick[1], Stephen A. Wood[1], Alex M. Sykes[1]***

**1** Griffith Institute for Drug Discovery, Griffith University, Nathan, Australia, **2** Queensland Brain Institute, University of Queensland, St Lucia, Australia

* a.sykes@griffith.edu.au

**Data Availability Statement:** Aligned whole exome sequencing data (BAM) presented in this report have been uploaded to NCBI SRA. BioProject ID:

## Abstract

The genetic study of multi-incident families is a powerful tool to investigate genetic contributions to the development of Parkinson's disease. In this study, we identified the rare *PTPRA* p.R223W variant as one of three putative genetic factors potentially contributing to disease in an Australian family with incomplete penetrance. Whole exome sequencing identified these mutations in three affected cousins. The rare *PTPRA* missense variant was predicted to be damaging and was absent from 3,842 alleles from PD cases. Overexpression of the wild-type RPTPα and R223W mutant in HEK293T cells identified that the R223W mutation did not impair RPTPα expression levels or alter its trafficking to the plasma membrane. The R223W mutation did alter proteolytic processing of RPTPα, resulting in the accumulation of a cleavage product. The mutation also resulted in decreased activation of Src family kinases. The functional consequences of this variant, either alone or in concert with the other identified genetic variants, highlights that even minor changes in normal cellular function may increase the risk of developing PD.

## Introduction

Parkinson's disease (PD) is the second most common neurodegenerative disease worldwide with prevalence estimated at 0.3% of the entire population, and 1% in persons over 60 years of age [1]. It is characterised by a slow and chronic loss of dopaminergic neurons in the substantia nigra pars compacta (SNpc), located in the midbrain [2]. PD presents primarily as motor symptoms, which include gate dysfunction, bradykinesia, rigidity, and resting tremors [3, 4].

The exact mechanism behind the pathogenesis of PD remains unknown. However, both environmental and genetic factors are suggested to play a role. Approximately 10–15% of PD patients report a first-degree relative with PD [5, 6], with *SNCA*, *LRRK2* and *VPS35* robustly linked to late-onset familial PD. Mutations in these genes are variably penetrant, but can lead to rare monogenic forms of PD, which account for approximately 2% of PD in populations of European descent [7, 8]. The identification of novel disease-causing mutations provides opportunities to understand aberrant molecular processes that can lead to disease. A genomic

PRJNA844215. SRA IDs: SRR19500741 and
SRR19500742.

**Funding:** Mellick GD. NHMRC project
APP1084560. The funders had no role in study
design, data collection and analysis, decision to
publish, or preparation of the manuscript.

**Competing interests:** The authors have declared
that no competing interests exist.

analysis of probands from 18 multi-incident families in Queensland had only identified one
kindred that could ascribe to a known genetic form of disease [9] suggesting that other con-
tributing factors have yet to be elucidated. Given both the discrepancy between cases with a
family history of disease and known monogenic carriers, as well as the benefit of identifying
novel targets for disease, we have been investigating novel genetic causes of PD using multi-
incident families.

In this study we identified affected members of a Queensland family with PD sharing three
missense single nucleotide variants (SNVs) in genes *PTPRA*, *ARL14EP* and *HYDIN* (Fig 1A
and 1B). Due to the change in charge and high Combined Annotation Dependent Deletion
(CADD) score for the p.R223W single nucleotide variation in *PTPRA*, this gene was chosen
for further analysis. *PTPRA* encodes the type I integral membrane phosphatase, receptor pro-
tein tyrosine phosphatase alpha (RPTPα). Mutations in *PTPRA* have been identified as segre-
gating with disease in a family with schizophrenia and in other unrelated patients with this
disease. Moreover, there is also evidence that variants in this gene may increase patients sus-
ceptibility to schizophrenia and autism spectrum disorders [10, 11]. RPTPα has a short extra-
cellular, N-glycosylated region, a transmembrane domain, a short cytosolic wedge region and
two catalytic phosphatase domains (D1 and D2) (Fig 1C). The catalytic activity of RPTPα is
found within this tandem arrangement of cytosolic phosphatase domains [12]. Unlike other
PTPases, both catalytic domains of RPTPα are active, though D1 (the membrane proximal
domain) has catalytic activity 4–5 orders of magnitude more active against phosphotyrosine
(pY) peptides than D2, the membrane distal domain [13]. RPTPα is responsible for activating
Src family kinases (SFKs) such as Src and Fyn by dephosphorylating their inhibitory phos-
phorylated tyrosine residues (Y527 and Y530 respectively) and promoting SFK signalling
for cell proliferation, integrin signalling and neuroinflammation [14–16]. The p.R223W
mutation is situated within the wedge region, just N-terminal of D1, in a region implicated in
homodimerisation stability [17] and activity regulation [18]. In addition to homodimerisation,
proteolytic cleavage has been suggested as a mechanism to regulate RPTPα activity. RPTPα
reportedly undergoes processing by calpain in the cytoplasm, resulting in reduced phosphatase
activity [19].

To study the functional consequences, if any, of RPTPα^R223W we generated mammalian
expression constructs and carried out systematic cellular and molecular approaches to evaluate
if RPTPα^R223W affected RPTPα activity.

## Methods

### Ethics statement

All donor tissue and information were obtained with informed and written consent of the
participants. All procedures were in accordance with National Health and Medical Research
Council Code of Practice for Human Experimentation and approved by the Griffith University
Human Experimentation Ethics Committee, Approval Number ESK/04/11/HREC.

### Genetic analysis

As detailed previously [9] the proband from family #431, a multi-incident family enrolled in
the Queensland Parkinson's Project, was found not to carry known and highly suspected
genetic causes of disease, including point mutations, indels, trinucleotide expansions and gene
dosage. An analysis of rare missense sequence variants shared with affected members of the
family was conducted through whole exome sequencing of the proband III:2, III:1 and III:4.
Briefly, Ion AmpliSeq exome libraries were sequenced on the Ion Proton (Thermo Fisher

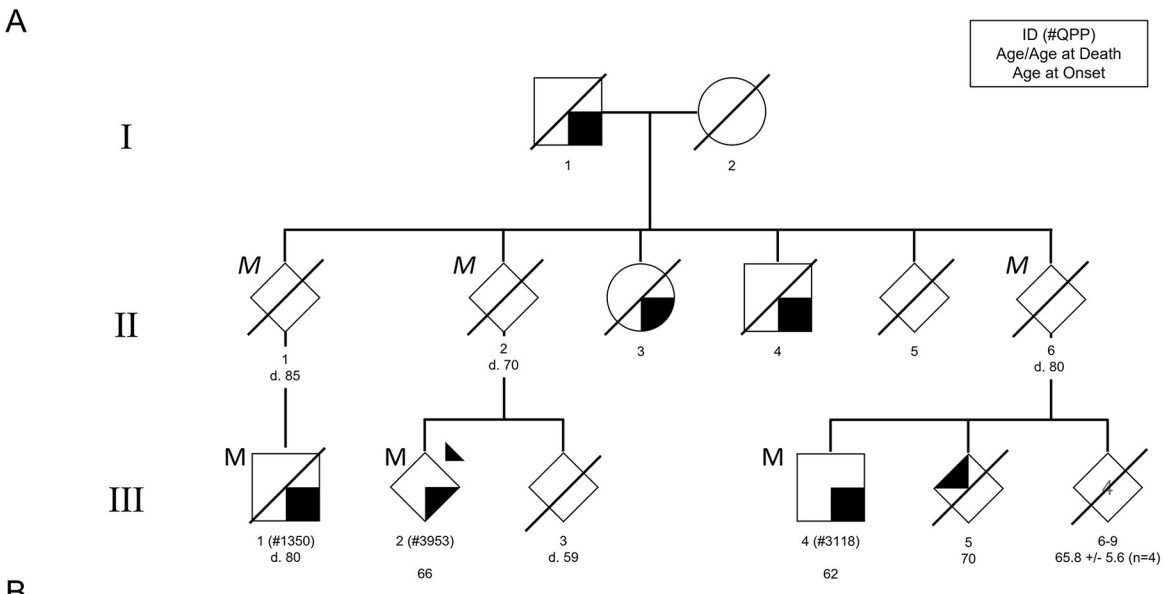

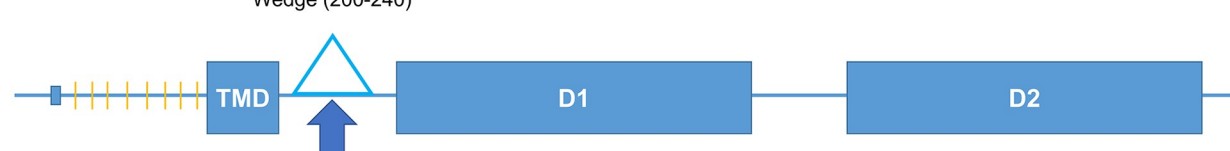

**Fig 1.** (A) Pedigree of the family carrying the *PTPRA* mutation. 'M' indicates confirmed heterozygous carriers of *PTPRA* mutation, italicised '*M*' indicates inferred genotype. Shaded lower right shapes represent PD and upper left represents undiagnosed tremor. Squares represent males, circles represent females and diamonds represent undisclosed gender. Dark triangle represents proband. Diagonal lines represent deceased members. Non-essential pedigree information has been omitted or modified to protect the privacy of the family. Subtext indicates pedigree ID (QPP ID), age / age of death and age at symptom onset. (B) Description of shared rare missense variants identified in three affected cousins. CDS: Coding DNA sequence, gnomAD: Genome Aggregation Database, SGC: Sydney Genomics Collaborative, CADD: Combined Annotation Dependent Depletion. (C) Schematic diagram of the RPTPα protein. Orange lines represent N-linked glycosylation sites. TMD = transmembrane domain. Arrow indicates the R223W mutation. D1 and D2 are the phosphatase domains.

Scientific), and subsequent variant calling was performed using the Torrent Suite (v4.0) (Thermo Fisher Scientific). Annotation was facilitated by the ANNOVAR package [20]. Sequence variants shared amongst the affected members and had a minor allele frequency less than 0.0001 in the gnomAD [21] were selected. Aligned whole exome sequencing data (BAM

## Expression constructs

RPTPα (splice variant 1; transcript ID ENST00000216877.10) was amplified from HEK293T cDNA using KOD polymerase with 5' *aat*B sites for gateway cloning, 5' Kozak sequence and no stop codon using the following primers: 5'-GGGGACAAGTTTGTACAAAAAAGCAGGCT TAGCCGCCAT GGATTCCTGGTTCATTCTTG-3' and 5'-GGGGACCACTTTGTACAAGA AAGCTGGGTACTTGAAGTT GGCATAATCTGAG-3'. RPTPα-*aat*B was subcloned into pDONR201 and then into pcDNA3.1-3xFLAG-V5-ccdB (a gift from Susan Lindquist & Mikko Taipale (Addgene plasmid #87064)) using BP Clonase II and LR Clonase II respectively according to the manufacturer's instructions (Thermo Fischer Scientific). Mutations were introduced by inverse PCR using KOD polymerase and the following primers for R223W: 5'- CTGGAAGAGGAAATTAACTGGAGAATGGCAGAC -3' and 5'- GTCTGCCATTCTC CAGTTAATTTCCTCTTCCAG -3', and for Sv4: 5'- CTTTCCCTCCTTCAGGTAATTCTGA CTCGAAGGACAGAAGAGATGAGACACCAATTATTG -3' and 5'-TTGGTGTCTATCTCTT CTGTCCTTCGAGTCAGAATTACCTGAAGGAGGGAAAGTTTC -3'. DpnI (NEB) treated mutagenesis PCR products were transformed into XL10-gold supercompetent *E. coli* (Agilent). Correct sequence identity was verified by Sanger sequencing.

## Cell culture and transfection

HEK293T cells (ATCC) were cultured in Dulbecco's modified Eagle medium (DMEM/F12; Thermo Fischer Scientific) supplemented with 10% foetal calf serum (FCS;Thermo Fischer Scientific) at 37˚C, 5% $CO_2$. For immunoblot experiments 100,000 cells were plated in a 24-well plate (Nunc) overnight and constructs transfected with Lipofectamine 2000 (Thermo Fischer Scientific). For immunofluorescence (IF) experiments, cells were plated overnight and transfected with Lipofectamine 2000. After 24 hours, cells were replated onto a 384 well CellCarrier plate (PerkinElmer). The cells were either lysed or fixed 48 hours post-transfection. Lysis was performed using TNE-TX buffer (10mM Tris, 150 mM NaCl, 1mM EDTA, 1% Triton X-100, cOmplete protease inhibitors (Sigma), cOmplete phosphatase inhibitors (Sigma)), incubating for 20 min on ice and centrifugation for 5 min at 4˚C. Alternatively, lysis for signalling experiments was performed using TNE-SDS buffer (10mM Tris, 150 mM NaCl, 1mM EDTA, 1% SDS, cOmplete protease inhibitors (Sigma), cOmplete phosphatase inhibitors (Sigma)), incubating 10 min on ice, before sonicating and centrifugation for 10 min. For IF experiments, cells were washed in ice cold PBS and fixed in 4% PFA (Sigma) for 10 min.

## Stable cells

pcDNA3.1-RPTPα-3xFLAG-V5, pcDNA3.1-RPTPα^R223W-3xFLAG-V5, pcDNA3.1-Sv4^WT-3xFLAG-V5 and pcDNA3.1-Sv4^R232W-3xFLAG-V5 were linearised with FspI and transfected into 6 well plates containing various densities of HEK293 cells. After 48 hours, cells were passaged in DMEM/F12 containing 150 μg/ml hygromycin (Thermo Fischer Scientific) and 10% FCS, with media replaced every three days until stable colony formation. All wells containing non-transfected HEK293 cells died in the presence of 150 μg/ml hygromycin. Stable colonies were isolated and expanded in DMEM/F12 containing 150 μg/ml hygromycin and 10% FCS and expression verified by immunofluorescence and immunoblotting.

## Cell treatments

For Endo H experiments lysates were aliquoted and treated in the presence or absence of 250U EndoH (NEB) for 2 hours at room temperature prior to immunoblotting. For cleavage experiment cells were treated 4 hours prior to lysis with, 10μM MG132 (Calbiochem), 1μM DAPT (Sigma), 200nM phorbol 12-myristate, 13-acetate (PMA, Sigma), 20μM TAPI-2 (Calbiochem), 10μM BB94 (Calbiochem), 10μM BIV (Calbiochem). For EGF experiments, stable RPTPα expressing HEK293 cells were plated at 500,000 cells/well in 6-well plates. Twenty-four hours after plating, the media was replaced with DMEM/F12 for 18 hours prior to addition of 100ng/ml EGF (Thermo Fischer Scientific) for the appropriate times prior to lysis.

## Immunoblotting

Samples were prepared in 1X SDS sample buffer ± 100mM DTT (BioRad), boiled for 5 minutes and centrifuged for 30 sec. Immunoblots were performed using Tris-Glycine gels electrophoresed and transferred to nitrocellulose (GE Healthcare) membranes using standard protocols. Membranes were blocked for 60 min in PBS containing either 3% non-fat milk powder or 5% BSA and probed with appropriate primary antibodies overnight at 4˚C (see Table 1. for details). Primary antibodies were detected by incubating the membranes in either: goat-anti-mouse-680RD and goat-anti-rabbit-800CW secondary antibodies (both 1:24,000); goat-anti-mouse-HRP antibody (1:20,000); or goat-anti-rabbit-HRP antibody (1:20,000) for 60 min. Membranes were imaged on an Odyssey-Fc imaging system (Licor).

## Immunoprecipitation

Protein G Sepharose 4 Fast Flow (GE Healthcare) beads were washed in TNE-TX before incubation with mouse anti-FLAG for one hour. The FLAG-sepharose was divided equally into tubes and each was incubated with lysates for 2 hours at room temperature by rotation. Immunoprecipitates were washed in TNE-TX and eluted in 2X SDS sample buffer, boiled for 5 mins and immunoblotted.

## DTSSP crosslinking

Two days post-transfection, cells were washed twice with cold PBS before incubation with cold 3,3′-Dithiobis(sulfosuccinimidylpropionate) (DTSSP) (Thermo Fischer Scientific) at 4˚C on ice for two hours. Excess DTSSP was quenched by the application of 1M Tris pH 7.5 for a further 15 minutes. Cells were then lysed using TNE-TX as previously described.

**Table 1. Antibodies used in this study.**

| Epitope | Company | Cat. No. | Dilution |
|---|---|---|---|
| FLAG | Sigma | F3165 | 1:3000 |
| V5 | Cell Signalling | #13202 | 1:3000 |
| α-tubulin | Sigma | T5168 | 1:12,000 |
| Phospho-PTPα (Y789) | Cell Signalling | #4481 | 1:1000 |
| Src (36D10) | Cell Signalling | #2109 | 1:3000 |
| Phospho-Src (Y416) | Cell Signalling | #6943 | 1:3000 |
| Non-phospho-Src (Y527) | Cell Signalling | #2107 | 1:3000 |
| Phospho-Tyrosine (P-Tyr-1000) | Cell Signalling | #8954 | 1:1000 |
| Phospho-ERK1/2 | Cell Signalling | #5726 | 1:3000 |
| Src (L4A1) | Cell Signalling | #2110 | 1:3000 |

## Microscopy

Fixed cells were blocked and permeabilised with PBS containing 1% BSA, 0.3% Triton X-100 for 60 mins at room temp and mouse-anti-FLAG (1:1000) were incubated for 90 min room temperature. After washing in PBS, detection was achieved by incubating with donkey-anti-mouse-488 (1:1000, Thermo Fischer Scientific) and DAPI (1:10,000, Sigma). Cells were imaged on an Operetta CLS system with an X63/1.15 objective.

## Statistical analysis

Statistical analysis was preformed using GraphPad Prism (v8). All data are presented as means ± standard deviation. Comparisons between two groups were analysed using Student's two-tailed unpaired $t$ test.

## Results

### Genetic analysis identified a PTPRA mutation shared among PD affected members of a Queensland family

The multi-incident family enrolled in the Queensland Parkinson's Project (QPP), family #431, presented with dominant inheritance, with notable incomplete penetrance, suggesting a substantial genetic component for disease within the family. The proband was excluded for known disease-causing mutations previously [9], thus analysis of the family aimed to identify putative disease-causing genetic lesions. The family presented with two unaffected members with inferred carrier status above the age of 80 years. Incomplete penetrance was anticipated due to the notable age- and ethnicity-dependent partial penetrance of the *LRRK2* p.G2019S [22]. Whole exome sequence analysis was performed in three affected cousins, III:1, III:2, III:4, which identified three rare heterozygous missense variants shared between the members, *PTPRA* p.R223W (NM_080840), *ARL14EP* p.A146V (NM_152316) and *HYDIN* p.A2271E (NM_001270974) (Fig 1A). These genes and variants have not been implicated in parkinsonism previously and presented as novel putative targets for disease. The variants in *PTPRA* and *ARL14EP* were predicted to be damaging by both the SIFT and PolyPhen-2 programs and had CADD scores of 28.6 and 21.5, respectively (Fig 1B). The *HYDIN* variant was not predicted to be damaging. As the ARL14EP$^{A146V}$ variant was a conservative replacement and in our initial studies we observed no obvious differences between the ARL14EP$^{WT}$ and ARL14EP$^{A146V}$ proteins in pilot immunoblot or subcellular localisation studies (unpublished) we decided to pursue the RPTPα protein variant further. Furthermore, the *PTPRA* p.R223W variant was absent in 3,842 PD alleles across seven countries from the GeoPD consortium (www.geopd.net), suggesting it is rare across multiple ethnicities.

### RPTPα$^{R223W}$ is transported to the plasma membrane

To investigate if the p.R223W mutation altered RPTPα protein stability and trafficking we first generated RPTPα$^{WT}$ and RPTPα$^{R223W}$ V5/Flag-tagged expression constructs. HEK293T cells were transfected with the RPTPα$^{WT}$ construct, lysed and analysed by immunoblot. RPTPα$^{WT}$ resolved primarily as two bands at ~160 kDa and ~120 kDa (Fig 2A). While the observed MW of RPTPα-V5/Flag is approximately 30 kDa larger than what is most commonly reported in the literature [23–26] it is not uncommon for the protein to be reported larger than 130 kDa [27–31] regardless of number or position of tags. As RPTPα undergoes N-glycosylation [23] the two bands likely represented post-Golgi (complex N-glycans) and endoplasmic reticulum (ER) forms of the protein. We confirmed this by treating lysates with endoglycosidase H (EndoH) which cleaves high mannose N-linked glycans present only in the ER and found that

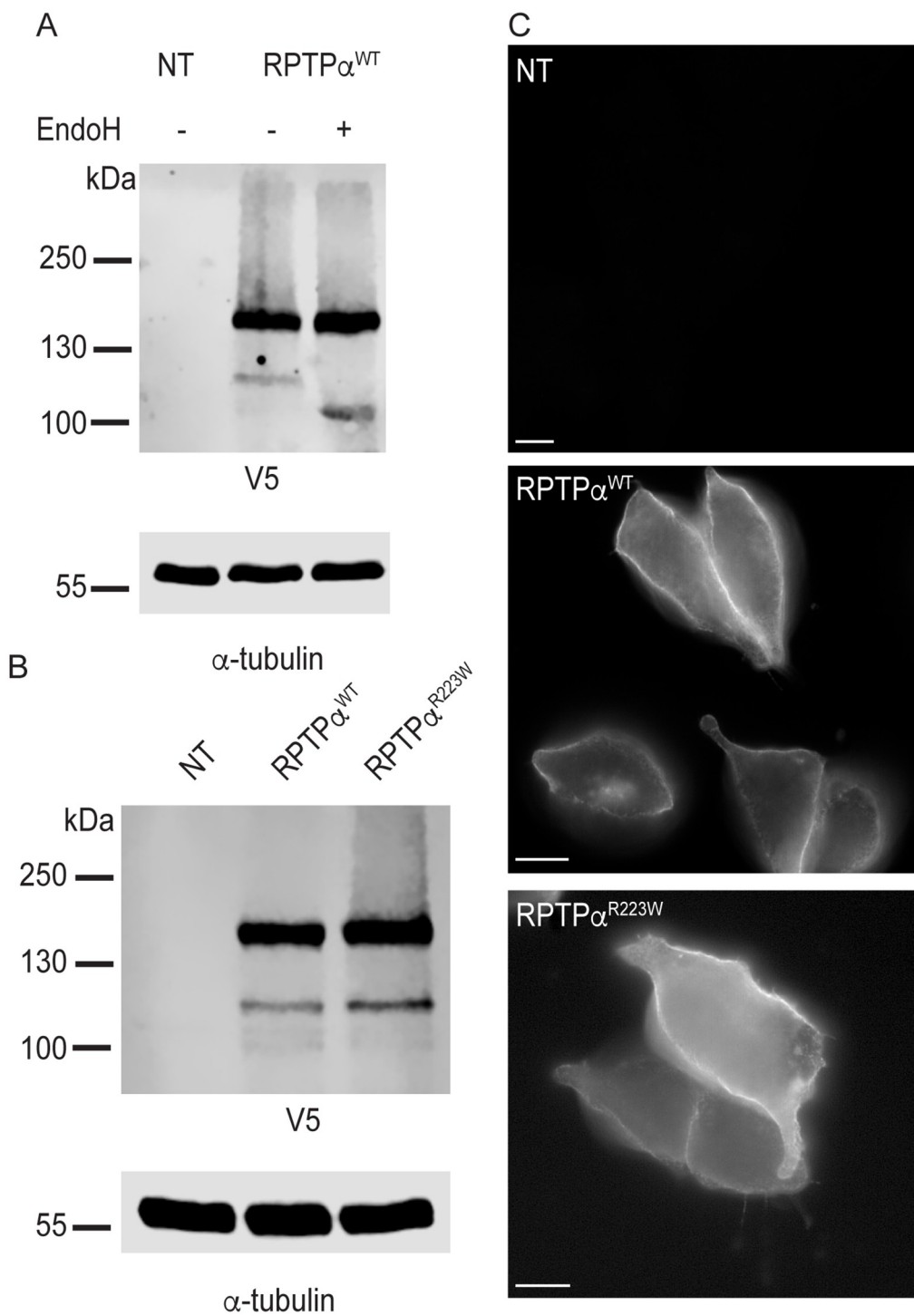

**Fig 2.** (A) Immunoblot of non-transfected (NT) and RPTPα wild-type (RPTPα$^{WT}$) HEK293T cells. The lysates were treated in the presence (+) or absence (-) of endoglycosidase H (EndoH) detected with anti-V5 antibody. (B) Immunoblot of RPTPα$^{WT}$ and RPTPα$^{R223W}$ transfected HEK293T cells detected with anti-V5 antibody. (C) Immunofluorescent image of transfected HEK293T cells stained with anti-FLAG antibody. Scale bar = 10μm.

the ~120 kDa band was indeed sensitive to EndoH treatment whereas the ~160 kDa band was insensitive (Fig 2A). Next, we expressed both RPTPα$^{WT}$ and RPTPα$^{R223W}$ in HEK293T cells, lysed and analysed by immunoblot to investigate if the R223W mutation altered RPTPα stability. We found that both RPTPα$^{WT}$ and RPTPα$^{R223W}$ resolved primarily as the post-Golgi ~160 kDa form, with no significant change in abundance (Fig 2B). We next performed immunocytochemical staining of both RPTPα$^{WT}$ and RPTPα$^{R223W}$ transfected HEK293T cells to investigate the subcellular localisation of both proteins. As expected, we found that both the WT and R223W proteins were localised primarily at the plasma membrane (Fig 2C) indicating that the R223W protein was indeed correctly trafficked to the plasma membrane.

## RPTPα$^{R223W}$ undergoes aberrant proteolytic cleavage

Upon further experimentation we discovered that RPTPα$^{WT}$ and RPTPα$^{R223W}$ also resolved as two extra lower molecular weight (MW) bands ~75 kDa (C1) and ~70 kDa (C2) (equivalent to ~70 kDa and ~65 kDa respectively without the C-terminal epitope tag) (Fig 3A) which fit with the literature for proteolytic cleavage, a common mechanism for regulation of type I membrane proteins, including RPTPα [19, 24, 32]. RPTPα is cleaved by an unknown protease to liberate the ~70kDa band [24] and by calpains to generate the ~65kDa band [19]. Surprisingly, we discovered that RPTPα$^{R223W}$ resolved a significantly higher proportion of the C1 band (Fig 3A) suggesting that proteolysis was perturbed in RPTPα$^{R223W}$ expressing cells. Based on the literature and the observed MW [24, 32] we hypothesised that the C1 band represented another metalloproteinase generated fragment and the C2 band was generated by calpain cleavage [19]. To validate that the C2 band was generated by calpains we treated RPTPα$^{R223W}$ expressing cells with MG132, which inhibits calpain cleavage of RPTPα [19]. As expected, the C2 band was not observed after MG132 treatment and the C1 fragment was not affected (Fig 3B). We also treated RPTPα$^{R223W}$ expressing cells with DAPT to investigate if the C1 fragment was produced by γ-secretase cleavage and discovered that neither cleavage fragment was altered by DAPT (Fig 3B). Following this, we treated RPTPα$^{R223W}$ expressing cells with metalloproteinase stimulator (PMA), metalloproteinase inhibitors (TAPI-2, BB94) and beta-secretase inhibitor IV (BIV), expecting that PMA would increase the abundance of the C1 band, TAPI-2/BB94 would inhibit the generation of the C1 band and BIV would have no effect on the banding pattern (Fig 3C). As expected BIV had no effect on the banding pattern. Unexpectedly, stimulation with PMA led to an increase in the C2 species and did not increase the abundance of the C1 fragment (Fig 3C). Furthermore, the broad-spectrum metalloproteinase inhibitors (TAPI-2 and BB94) inhibited the generation of the C2 band. These inhibitors had no influence on the C1 band (Fig 3C) indicating that the C2 species is a generated by metalloproteinases prior to calpain cleavage, and is not reliant on the production of C1. The C1 band is a cleavage product generated by an unknown mechanism that is independent of calpain cleavage. This cleavage product is significantly more abundant in RPTPα$^{R223W}$ expressing cells. Interestingly, when we investigated the phosphorylation state of both RPTPα$^{WT}$ and RPTPα$^{R223W}$ there was no difference in either phospho-RPTPα$^{Y789}$ (Fig 3D) or total phosphotyrosine of the full-length protein (Fig 3E). Furthermore, when we investigated the tyrosine phosphorylation status of the C1 and C2 fragments we observed that they were not phosphorylated (Fig 3C and 3D) indicating that these fragments are potentially catalytically inactive.

## The RPTPα$^{R223W}$ mutation does not affect phosphatase function but does alter SFK activation

As the R223W mutation lies within the wedge region, involved in homodimerisation, we expected that homodimerisation would be altered in RPTPα$^{R223W}$ expressing cells. RPTPα

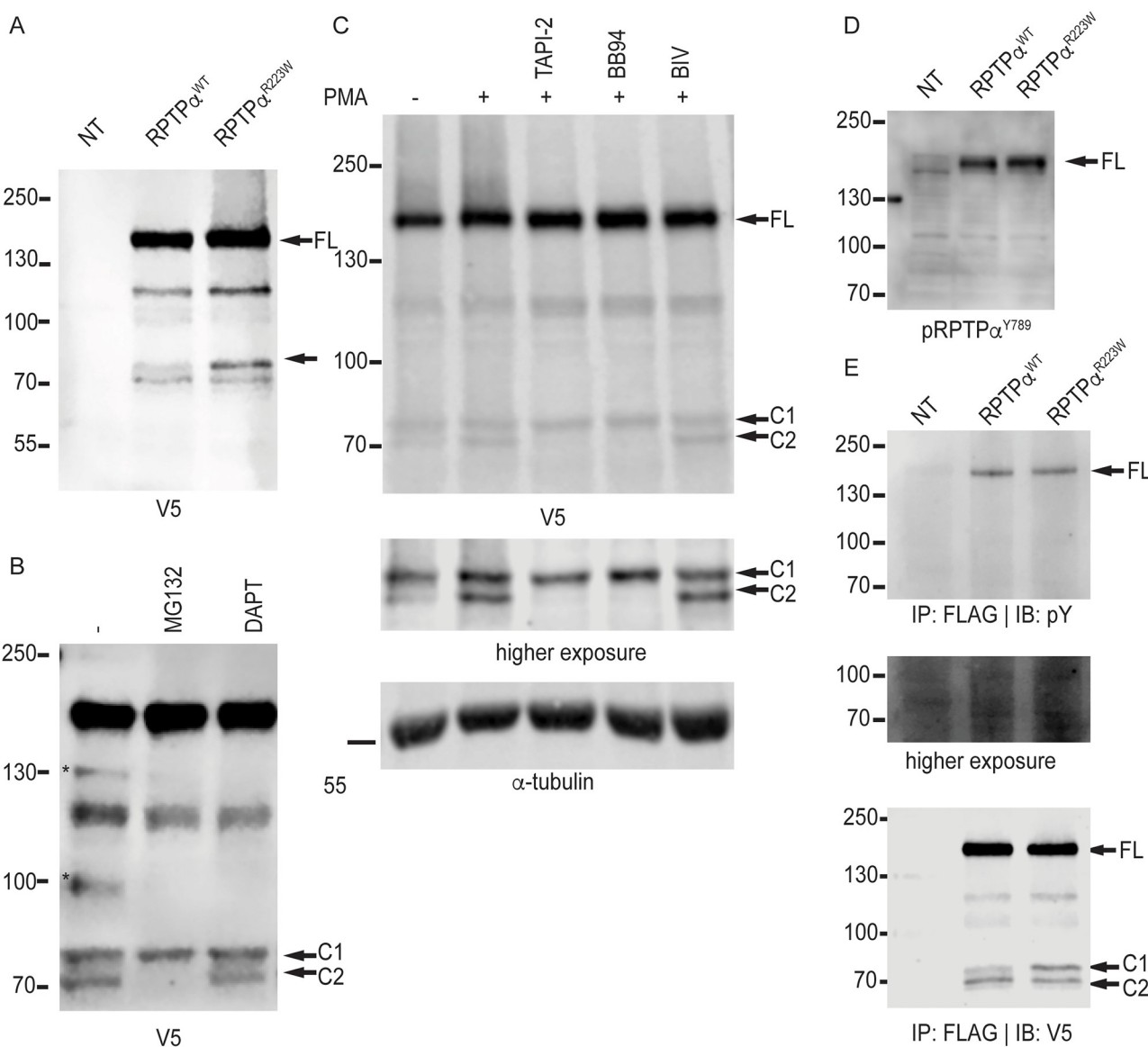

**Fig 3.** (A) Immunoblot of RPTPα^WT and RPTPα^R223W transfected HEK293T cells. Arrow indicates RPTPα^R223W specific ~75 kDa band. (B) Immunoblot of RPTPα^R223W transfected HEK293T cells treated with MG132 or DAPT for 4 hours. (C) Immunoblot of RPTPα^R223W transfected HEK293T cells treated with PMA, TAPI-2, BB94 or BIV for 4 hours. (D) Immunoblot of RPTPα^WT and RPTPα^R223W transfected HEK293T cells probed with pRPTPα^Y789 antibody. (E) Immunoblot of RPTPα^WT and RPTPα^R223W transfected HEK293T cells subjected to immunoprecipitation with FLAG antibody and probed with either phospho-tyrosine (pY) or V5 antibodies.

dimerisation is a transient event that is difficult to detect under steady state conditions requiring small molecule cross linkers for detection [25, 26]. Thus, we employed DTSSP, a thiol reducible amine crosslinker, for homodimerisation studies between RPTPα^WT and RPTPα^R223W expressing cells. As expected, a small fraction of RPTPα^WT was recovered at a higher MW band corresponding to a dimer in the absence of DTT. Furthermore, these RPTPα dimers were confirmed by the absence of the higher MW band when the crosslinker was reduced by DTT (Fig 4A). Surprisingly, when we investigated RPTPα^R223W expressing cells, dimers were recovered at an equivalent amount as in RPTPα^WT expressing cells, suggesting that homodimerisation is not perturbed by the R223W mutation.

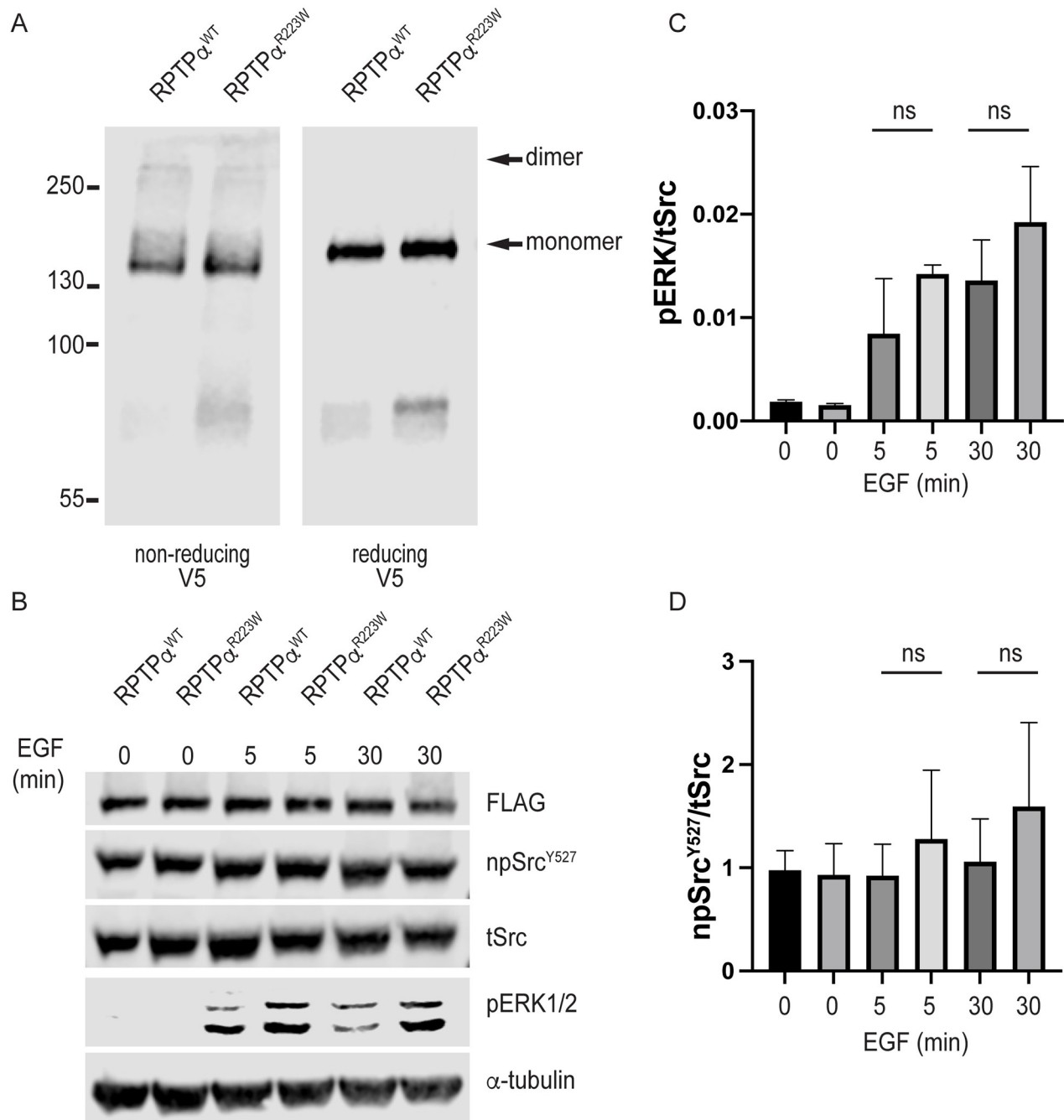

**Fig 4.** (A) Immunoblots of RPTPα transfected HEK293T cells under non-reducing or reducing conditions. Cells were crosslinked with DTSSP prior to lysis. (B) Immunoblots of HEK293 stable cells expressing RPTPα$^{WT}$ and RPTPα$^{R223W}$ treated with 100ng/ml EGF for the indicated times and probed with the labelled antibodies. (C) Quantification of the levels of pERK 1/2 (mean±SD, n = 3). (D) Quantification of the levels of npSrc$^{Y527}$ (mean±SD, n = 3).

As RPTPα acts directly on SFKs by dephosphorylating the inhibitory phosphorylated tyrosine (Src$^{Y527}$/Fyn$^{Y530}$) we next investigated if the RPTPα$^{R223W}$ mutation had altered activity against these phosphorylated residues. To examine this, we generated stable HEK293 lines expressing RPTPα$^{WT}$ and RPTPα$^{R223W}$. We depleted FCS from the RPTPα expressing cells for

18 hours and treated them in the presence and absence of 100 ng/ml EGF to reinitiate EGFR signalling [33]. We then lysed the cells and analysed the RPTPα activity by immunoblot analysis. As expected EGF stimulation resulted in ERK1/2 phosphorylation (Fig 4B). We observed a trend in increased ERK1/2 phosphorylation in RPTPα$^{R223W}$ cells compared to RPTPα$^{WT}$ cells (Fig 4C), though this increase did not reach significance. In addition, a trend toward increased phosphorylation of Src$^{Y527}$ in RPTPα$^{R223W}$ compared to RPTPα$^{WT}$ was observed (Fig 4D), though again did not reach significance. This data suggests that phosphatase ability of the R223W mutant is not significantly impaired compared to WT when signalling is stimulated by EGF.

To investigate if the potentially reduced dephosphorylation of Src$^{Y527}$ resulted in altered activation of Src, we next investigated the phosphorylation of Src$^{Y416}$, which undergoes autophosphorylation after the dephosphorylation of Y527. Stable HEK293 lines were depleted of FCS for 18 hours and then lysed and analysed by immunoblot. Cells were not treated with EGF as we found no added effect on Y416 above that seen with serum starvation (S1 Fig). In this paradigm we found no significant decrease in the ratio of Src$^{Y416}$ phosphorylation in the RPTPα$^{R223W}$ expressing cells (1.23±0.07) compared to RPTPα$^{WT}$ (1.45±0.23) (Fig 5). To elucidate if these results were unique to the short RPTPα isoform, we repeated the experiment in HEK293 cells stably expressing the longer splice variant: RPTPα-Sv4$^{WT}$ and RPTPα-Sv4$^{R232W}$. In contrast to splice variant 1, we observed a significant decrease in Y416 phosphorylation in RPTPα-Sv4$^{R232W}$ cells (1.03±0.15) compared to RPTPα-Sv4$^{WT}$ (1.50±0.19) expressing cells. These results suggest that while phosphatase activity is not affected by the R223W/R232W mutation downstream activation of Src is reduced in RPTPα-Sv4.

## Discussion

In this study we identified a rare, functional *PTPRA* variant that segregates with parkinsonism in a multiplex family. We propose that the presence of this functional mutation may increase risk of development of parkinsonism in concert with the other identified variants and environmental factors. Interestingly, rare variants in *PTPRA* were also recently found in a family study of schizophrenia, though their functional significance has not been elucidated [10]. While this variant is rare, with a minor allele frequency of 5.34 x $10^{-5}$, it is more common than the *LRRK2* p.R1441H/G/C, *VPS35* p.D620N or the *SNCA* p.A53T or p.A30P variants [21].

While parkinsonism status of participants of gnomAD is not disclosed, it would be mistaken to assume that all *PTPRA* p.R223W carriers develop PD. Rather, we hypothesise that in this family, additive to their other genetic and environmental risk factors, the perturbed activity of the protein due to this variant conferred risk for developing PD before the age of 80 years. To examine the hypothesis that the genetic change in isolation increases risk of disease, by 5-fold with 80% power, would require a balanced case-control study of 120,000 participants. However, it is much more challenging to examine the risk in the context of this particular family and will require follow-up assessments of the next generation and further characterisation of the role of the aberrant proteins. Recently, during the submission process a new PD database was published (https://pdgenetics.shinyapps.io/VariantBrowser/), which includes new sequence data from the International Parkinson Disease Genomics Consortium (IPDGC) and the United Kingdom Biobank (UKB). The *PTPRA$^{R223W}$* variant was identified in 4/5141 PD patients and 42/42754 controls. Both the *ARL14EP* and *HYDIN* variants were absent. The UKB has collected extensive phenotypic data about their participants through surveys and imaging tools, however, there are still noteworthy constraints on the data: (1) only 10% of participants are over 80 years of age. (2) 3,602 participants had PD, however, 19,460 parents of participants were reported to have PD [34, 35]. Carrier status of these participants was

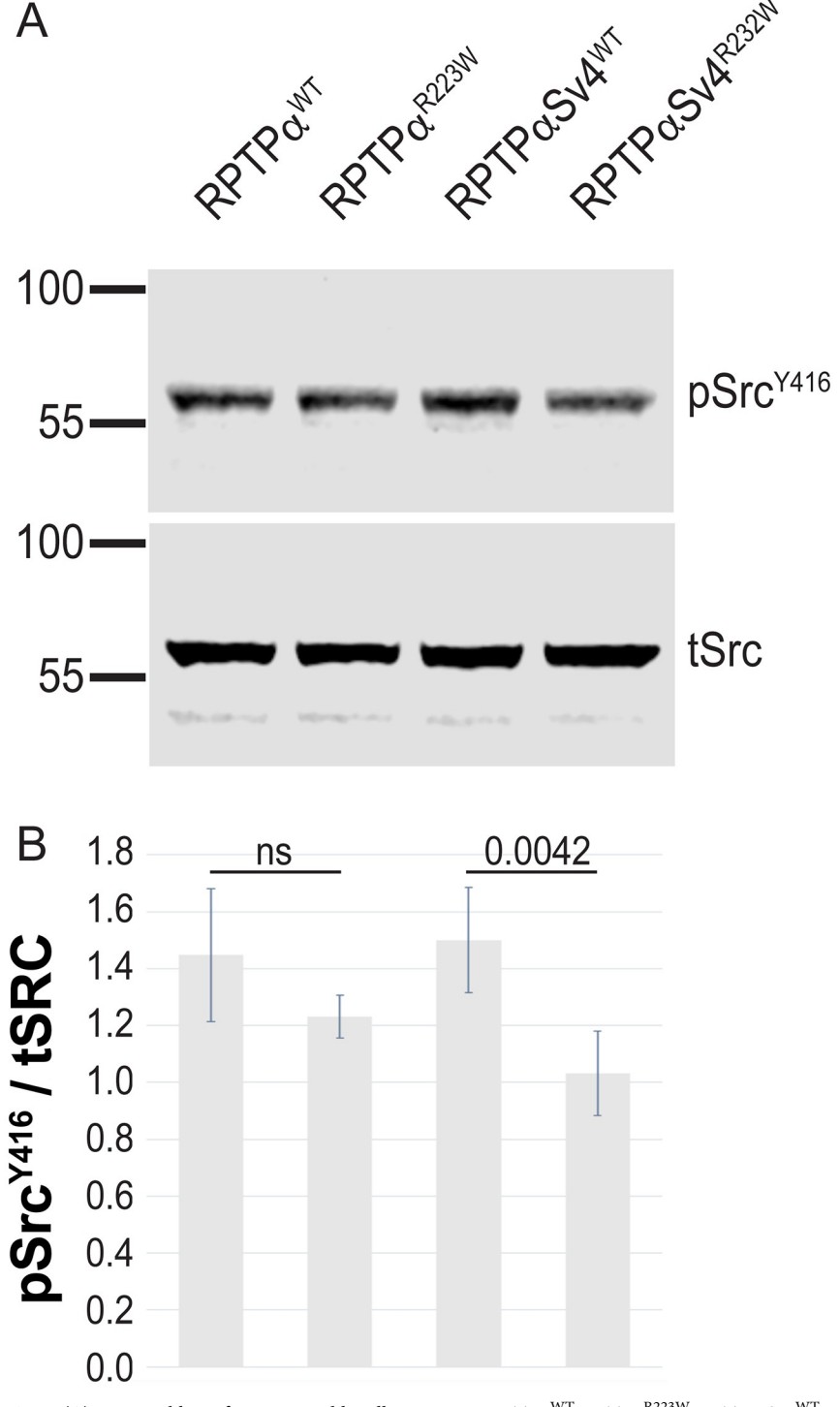

**Fig 5.** (A) Immunoblots of HEK293 stable cells expressing RPTPα$^{WT}$, RPTPα$^{R223W}$, RPTPα-Sv4$^{WT}$ and RPTPα-Sv4$^{R232W}$ and probed with the labelled antibodies. (B) Quantification of the ratio of pSrc$^{Y416}$ signal in RPTPα mutants compared to their respective RPTPα$^{WT}$ (mean±SD, n = 5).

unknown. (3) The dataset had disclosed 30.3% of participants were related up to the third-degree [34], indicating frequency of rare variants may be exaggerated due to these familial clusters. Given the notable incomplete penetrance of known PD genes by the age of 80 years [22] and given the absence of extended clinical family history, relatedness and age with these 42 controls, it is still plausible that the *PTPRA* SNV we identified may have a role in PD. Further, this could be further exacerbated by other rare variants, such as the *ARL14EP* and *HYDIN* SNVs and/or other genetic background and environmental factors. Nonetheless, the *PTPRA*$^{R223W}$ variant had altered the normal protein function. In the ectopic overexpression experiments presented here, the RPTPα$^{R223W}$ protein appeared functional and was transported to the plasma membrane correctly suggesting that this SNV does not render the RPTPα protein unstable. Even though the R223W mutation was located within the wedge region responsible for stability of dimerisation, no difference in dimerisation was observed compared to the wild-type RPTPα. However, the R223W expressing cells did display a significant increase in one proteolytic cleavage product, in addition to reduced activation of Src for the longer splice variant, despite apparently unchanged phosphatase ability of the mutant.

It was unexpected that the cytoplasmic located R223W mutation would influence proteolytic cleavage. However, given the size of the substituted amino acid, it is possible that enough of a change in protein structure was introduced to facilitate enhanced proteolytic cleavage. Even though we were unable to determine the exact mechanism by which this cleavage occurred, we can conclude that metalloproteinases, β-secretase, γ-secretase and calpains are not the mechanism for the enhanced R223W cleavage fragment. Given that we utilised a C-terminal epitope tag for detection of the cleavage fragment we can conclude that this fragment is equivalent to the 68 kDa fragment observed but not studied further by Kapp et al. [24].

Typically, PD is the result of progressive loss of dopaminergic neurons with age, therefore we would not expect a PD-causing mutation in *PTPRA*, a gene important for neuronal development [36], to have a strong/severe phenotype as this would result in many other neurological problems. Although dimerisation was unaffected in this overexpression paradigm, the change in RPTPα$^{R223W}$ proteolytic processing might result in the generation of a dimer inactivating cleavage fragment that results in altered phosphatase activity similar to inactivation of receptor tyrosine kinases by proteolytic cleavage [37]. The absence of phosphorylation at Y789 in these cleaved fragments suggests they may be catalytically inactive against SFK substrates [38–40]. According to the tyrosine displacement model presented by Zheng and collegues [38], pY789 of RPTPα is required to displace the Src SH2 domain, allowing access to pY527. However, several studies dispute this model, showing that Tyr789 phosphorylation is not necessary for pY527 dephosphorylation [31, 41–43]. Meanwhile it has reported that activity of the Y789F mutant depended on substrate, with a pool of mutants still active against EGFR/Src, and PTPRA-potentiated ERK activation unaffected by Y789F mutants [33]. As no consensus has yet been reached as to the necessity of pY789, further work is required to verify the activity of C1 and C2.

The ability of RPTPα to dephosphorylate Src$^{Y527}$ was unaffected by the R223W/R232W mutation. However, Src$^{Y416}$ phosphorylation was perturbed in R232W expressing cells. Previous studies have shown that while RPTPα induces the activation of Src kinase by reducing the phosphorylation of Y527, a reduction in overall Src kinase phosphotyrosine levels was also observed [27]. This suggests that either Y527 dephosphorylation was not followed by Y416 phosphorylation, or that RPTPα is also capable of dephosphorylating Src$^{Y416}$ [38, 40]. In unstimulated thymocytes RPTPα actually had an overall negative effect on Fyn activation, due to its ability to dephosphorylate the Fyn equivalents of Y527 and Y416 [44]. Perhaps the R232W variant slows down the disassociation of RPTPα from the SH2 domain of Src, providing physical impairment to Src$^{Y416}$ auto-phosphorylation. The region the mutation resides in is important

for the formation and stabilisation of dimers [18, 45] and is predicted to be important for protein-protein interactions [46]. The D1 domain is important for the formation of the active site pocket [45], but the upstream wedge region where R223 resides has no defined contribution to phosphatase activity. However, elucidation of the detailed mechanism underlying the reduction in Src$^{Y416}$ phosphorylation in the R223W/R232W variants is beyond the scope of the present study.

Liberation of the RPTPα ectodomain may also play a role in the regulation of SFK signalling by either stabilising or inhibiting RPTPα dimerisation or activating other membrane proteins, as for amyloid precursor protein (APP) processing. The β-secretase BACE1 cleaves APP at the N-terminus of the Aβ domain, which is the first step in the formation of pathogenic Aβ in Alzheimer's disease [47]. Conversely, α-secretase can cleave within the Aβ domain, thereby precluding Aβ generation and producing a fragment (sAPPα) that is believed to be neuroprotective [47]. sAPPα is believed to promote neurite outgrowth, synaptogenesis and cell adhesion [48, 49] while APPβ acts as ligand for DR6, promoting caspase 6 activation resulting in axonal pruning [50].

This study has identified a novel SNV linked to familial PD. Two other SNVs were also shared amongst affected members of the multi-incident family; these may also be involved or required for PD progression and further studies into these mutations are required to understand their role in PD. Nonetheless, this putative PD associated mutation, *PTPRA* p.R223W/R232W does generate a functional protein which is cleaved by an unknown protease to a greater extent than the wild-type protein and results in impaired Src activation in R232W expressing cells. Whether this contributes to increased risk of PD alone, or in concert with the other SNVs, still remains to be determined.

## Supporting information

**S1 Fig. Original uncropped western blots.**
(PDF)

**S1 Raw images.**
(DOCX)

## Acknowledgments

The main funding for this project comes from NHMRC project APP1084560: Identification of Parkinson's disease genes in Queensland families showing patterns of Mendelian inheritance. The GeoPD consortium whom had provided samples for genotyping. The patients, families and staff members involved in the Queensland Parkinson's Project.

## Author Contributions

**Conceptualization:** Melissa A. Hill, Steven R. Bentley, Tara L. Walker, George D. Mellick, Stephen A. Wood, Alex M. Sykes.

**Data curation:** Melissa A. Hill, Steven R. Bentley, Alex M. Sykes.

**Formal analysis:** Melissa A. Hill, Alex M. Sykes.

**Funding acquisition:** George D. Mellick.

**Investigation:** Melissa A. Hill, Steven R. Bentley, George D. Mellick, Stephen A. Wood, Alex M. Sykes.

**Methodology:** Melissa A. Hill, Steven R. Bentley, Tara L. Walker, Stephen A. Wood, Alex M. Sykes.

**Project administration:** George D. Mellick, Stephen A. Wood, Alex M. Sykes.

**Resources:** George D. Mellick.

**Supervision:** George D. Mellick, Stephen A. Wood, Alex M. Sykes.

**Validation:** Melissa A. Hill, Steven R. Bentley, Alex M. Sykes.

**Visualization:** Melissa A. Hill, Alex M. Sykes.

**Writing – original draft:** Melissa A. Hill, Steven R. Bentley, George D. Mellick, Stephen A. Wood, Alex M. Sykes.

**Writing – review & editing:** Melissa A. Hill, Steven R. Bentley, Tara L. Walker, George D. Mellick, Stephen A. Wood, Alex M. Sykes.

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
