## [Decision Letter · Decision Letter 0]

18 Mar 2021

PONE-D-21-04758

Does a rare mutation in PTPRA contribute to the development of Parkinson’s disease in an Australian multi-incident family?

PLOS ONE

Dear Dr. Sykes,

Thank you for submitting your manuscript to PLOS ONE. After careful consideration, we feel that it has merit but does not fully meet PLOS ONE’s publication criteria as it currently stands. Therefore, we invite you to submit a revised version of the manuscript that addresses the points raised during the review process.

Both reviewers noted issues with the manuscript and Reviewer 1 suggested rejection.  We would be willing to consider a revised manuscript, but these issues would have to be addressed.

We look forward to receiving your revised manuscript.

Kind regards,

Salvatore V Pizzo

Academic Editor

PLOS ONE

Journal Requirements:

Reviewers' comments:

Reviewer's Responses to Questions

**Comments to the Author**

1. Is the manuscript technically sound, and do the data support the conclusions?

Reviewer #1: Partly

Reviewer #2: Partly

2. Has the statistical analysis been performed appropriately and rigorously? 

Reviewer #1: I Don't Know

Reviewer #2: Yes

3. Have the authors made all data underlying the findings in their manuscript fully available?

Reviewer #1: Yes

Reviewer #2: Yes

4. Is the manuscript presented in an intelligible fashion and written in standard English?

Reviewer #1: Yes

Reviewer #2: Yes

5. Review Comments to the Author

Reviewer #1: In their manuscript, Hill et al. investigated a rare mutation of the receptor type protein tyrosine phosphatase RPTPα. They discovered the mutant in a family with Parkinson's disease (PD) and localized it to the wedge region in the intracellular domain. Physiological characterization of the mutant showed intact processing and trafficking but altered proteolytic processing and a delayed catalytic activty. However, a causal role for PD remained speculative.

The study in principle is of interest. However, even as an initial study, it has several problems. First, how likely is a causal role of RPTPα for the development of Parkinson's disease when the new sequence data (page 12) show a similar prevalence of the mutant in patients and controls (~1:1000)?

Next, the cDNA of RPTPα was newly cloned. How good was the sequence characterized? The transiently expressed protein shows an abnormal molecular weight. RPTPα is normally found as a protein of 130 kD and not 180. The proteolytic processing of the extracellular domain of RPTPα occurs only for a splice variant containing an additional 9 amino acids in the extracellular domain. Has this variant been cloned? If so, why then do we not see a total of three processed fragments, as has been published: 75 kD for the membrane anchored form, and 63 and 68 kD for the cytoplasmic products of calpain cleavage? Why is there only little stimulation of the metalloprotease by PMA (Fig. 3B)? And why was the metalloprotease not stimulated with POV which works better? Similarly, the calpain protease was not tested by stimulation with a Ca-ionophore. Since the authors claim the 70 kD protein fragment to be the product of the metalloprotease, this would have been essential. Finally, the term "regulated intracellular proteolysis" (p10) is inappropriate for an extracellularly cleaving metalloprotease.

The immunofluorescence shown in Fig. 2C is of very low quality and does not allow to see a difference between overexpressing and background cells, making it difficult to accept that trafficking of the mutant protein is normal. The phalloidin staining is a complete failure.

The authors find no tyrosine phosphorylation on the proteolytically processed fragments and conclude that these do not possess catalytic activity. However, they do not prove and not even give a reason for their hypothesis. In addition, the authors provided the same image in Fig. 3A and 3D. This becomes obvious when looking at the complete blot in the supplemental figure.

In Fig. 4A, dimerization of the mutant phosphatase is tested. As a control, a delta-wedge mutant is added. This protein is hardly expressed which should at least be discussed.

Finally, in Fig. 4B the effect of the mutant on Src activation is investigated. Phosphorylation of Src-Y527 is more strongly present 10 min after serum stimulation than in presence of the wild type form. This is a surprising result but it is also surprising that the inhibitory Src-Y527 phosphorylation increased at all after serum treatment for both phosphatase variants. The considerably varying levels of phosphatase expression may be one explanation, however, also the level of activating phosphorylation of Akt is higher before stimulation than after 5 and 10 min. This rather points to a general experimental problem. It would have been less error prone to establish cell lines overexpressing RPTPα than performing transient expression experiments. Last, the tubulin blot in Fig. 4B, right panel is part of the total blot shown for RPTPα in the supplemental figure. How can it appear as a separate blot without additional bands as the last figure of the supplemental data?

Reviewer #2: Hill et all identified three candidate genes that might contribute to Parkinson’s disease in an Australian family. Of these, the PTPRA p.R223W variant was the most likely to contribute to the disease. The R223W mutation did not affect the expression level of RPTPalpha and RPTPalpha was still expressed on the cell membrane. However, the R223W mutation led to the formation of a novel proteolytic cleavage product. The R223W mutation did not affect dimerization of RPTPalpha. Yet, the dynamics of Src Tyr527 phosphorylation appeared to be different in cells expressing the R223W variant, compared to wild type RPTPalpha. The authors conclude that a rare familial mutation in PTPRA alters its proteolytic processing and activity and may contribute to the cause of Parkinson’s disease.

This is an interesting report which shows that signaling of RPTPalpha is affected by R223W mutation. The correlation of the R223W mutation with Parkinson’s disease is not fully penetrant, which may be due to genetic enhancers and/or suppressors of the phenotype.

Points:

1. Src Tyr527 phosphorylation appears to be affected by expression of the R223W variant, in that serum stimulation results in a transient increase in Tyr527 phosphorylation for about 30 min in the presence of wild type RPTPalpha and only 10 min in the presence of the R223W variant (Fig.4). The authors claim that Src activity is affected differently by wild type and mutant RPTPalpha. This is not evident from the data provided. Tyr527 is an inhibitory phosphorylation site, which is phosphorylated by Csk. Activation of Src will lead to enhanced phosphorylation of the autophosphorylation site, Tyr416. Phosphorylation of Tyr416 is generally taken as a measure for Src activity. The authors should therefore reprobe their blots with Src pTyr416-specific antibodies to be able to conclude that Src activity is affected. It is noteworthy that both pTyr416 and pTyr527 are substrates of RPTPalpha.

Minor points:

1. The last sentence of the main text does not read well: “This results in a subtle, yet significant delay in activity potentially altering SFK signalling would be altered in the brains of affected patients.”

2. Ref 9 appears to be the same as ref 20.

6. PLOS authors have the option to publish the peer review history of their article (what does this mean?). If published, this will include your full peer review and any attached files.

Reviewer #1: No

Reviewer #2: No

---

## [Author Response · Author response to Decision Letter 0]

13 Jul 2021

Reviewer #1: 

The study in principle is of interest. However, even as an initial study, it has several problems. First, how likely is a causal role of RPTPα for the development of Parkinson's disease when the new sequence data (page 12) show a similar prevalence of the mutant in patients and controls (~1:1000)?

The reviewer raises a valid question and which we had considered carefully before composing the manuscript. While the International Parkinson’s Disease Genomics Consortium and United Kingdom Biobank data does suggest similar prevalence rates, we describe the limitations of this data. Referring specifically in the discussion, “(1) only 10% of participants are over 80 years of age. (2) 3,602 participants had PD, however,19,460 parents of participants were reported to have PD (Bycroft et al., 2018; Van Hout et al., 2020). Carrier status of these participants was unknown. (3) The dataset had disclosed 30.3% of participants were related up to the third-degree (Bycroft et al., 2018), indicating frequency of rare variants may be exaggerated due to these familial clusters”, the age of participants from the UKB, unknown carrier status of participants with PD affected relatives and the high rate of third-degree or closer relationships decreases the reliability of this dataset as a true measure of the rate of the rare mutation in an unaffected control population. Nevertheless, we have included this data for transparency. Furthermore, we also propose additional factors, such as the other 2 rare mutations, other genetic or environmental factors, may operate in synergy with the aberrant RPTPα functionality to increase disposition of disease, which increases the complexity of the variant-disease relationship. Referring to the discussion text: which has been adjusted to now read “Further, this could be further exacerbated by other rare variants, such as the ARL14EP and HYDIN SNVs and/or other genetic background and environmental factors.“

We also describe in the discussion, to assess the risk of this variant in isolation, we would require a balanced case-control study of 120,000 participants, which unfortunately, is larger than currently available IPDGC or UKB datasets or feasible for us to perform.

Next, the cDNA of RPTPα was newly cloned. How good was the sequence characterized? The transiently expressed protein shows an abnormal molecular weight. RPTPα is normally found as a protein of 130 kD and not 180. 

In our study we have used RPTPα SV1(793AA protein, transcript ID ENST00000216877.10, also termed PTPα123 (Kapp et al. 2012)) and not RPTPα SV4 (802AA protein, transcript ID ENST00000399903.7, also termed PTPα132 (Kapp et al. 2012)). We have fully sequenced the open reading frame and confirmed that it is correct. Furthermore, the observed MW of the mature full length RPTPα SV1 @~160kDa (between the 250 and 130 kDa bands) is consistent with (Yao et al. 2017) which studied RPTPα SV1 – as designated by the Y789F mutation (Y798 in SV4) Fig 6E and supplemental figure S5 A, B – PTPRA-FLAG migrating between 250/130 kDa and the “precursor” ~120 kDa which is equivalent to what we also observe in our study. 

To make it absolutely clear which splice variant has been used the materials and methods now reads “RPTP⍺ (splice variant 1; transcript ID ENST00000216877.10) was amplified from HEK293T cDNA”

The proteolytic processing of the extracellular domain of RPTPα occurs only for a splice variant containing an additional 9 amino acids in the extracellular domain. 

We initially believed the bands to be produced by RIP and after reanalysing the data have completely rewritten this section. Aside from our study, RPTPα SV1 cleavage bands also present @ 68 and 63 kDa Fig 1A (Kapp et al. 2012). In this paper the 63 kDa band is interoperated to be produced by calpain cleavage and the 68 kDa “has the size of the PTP�intra and might occur right at the plasma membrane” was not studied further. 

In our hands the equivalent bands at ~75kDa (C1), which may be cleaved close/at the membrane is not stimulated by PMA or inhibited by TAPI2/BB94 indicating that it is not generated by metalloproteinase. However, the ~70kDa (C2) band, is stimulated by PMA and is completely inhibited by TAPI2/BB94 indicating that although it is generated by calpain cleavage it first requires cleavage by a metalloproteinase. 

Has this variant been cloned? If so, why then do we not see a total of three processed fragments, as has been published: 75 kD for the membrane anchored form, and 63 and 68 kD for the cytoplasmic products of calpain cleavage? Why is there only little stimulation of the metalloprotease by PMA (Fig. 3B)? And why was the metalloprotease not stimulated with POV which works better? Similarly, the calpain protease was not tested by stimulation with a Ca-ionophore. Since the authors claim the 70 kD protein fragment to be the product of the metalloprotease, this would have been essential. 

We decided to use PMA as it stimulated equivalent if not better cleavage of RPTPα SV4 (Kapp et al. 2012 Fig 1C and Fig 2A, termed TPA). Our hypothesis was that the R223W fragment was being generated by metalloproteinases and that PMA would increase its abundance. We have performed MG132 treatment and conclude that the 70kDa (C2) fragment is produced by calpain cleavage as its abundance is decreased by MG132 treatment (Figure 3B). However, PMA stimulates the generation of the 70 kDa (C2) band and this generation is completely blocked by TAPI-2, indicating an initial metalloproteinase step happens prior to calpain cleavage.

Finally, the term "regulated intracellular proteolysis" (p10) is inappropriate for an extracellularly cleaving metalloprotease.

We thank the reviewer for their comment and we agree and have adjusted the whole section accordingly using the terminology C1 and C2 for the two cleavage fragments. 

The immunofluorescence shown in Fig. 2C is of very low quality and does not allow to see a difference between overexpressing and background cells, making it difficult to accept that trafficking of the mutant protein is normal. The phalloidin staining is a complete failure.

We have completed new imaging with just anti-FLAG, with higher quality images. A non-transfected cell control is added to demonstrate the specificity of the antibody staining.

The authors find no tyrosine phosphorylation on the proteolytically processed fragments and conclude that these do not possess catalytic activity. However, they do not prove and not even give a reason for their hypothesis. 

We thank the reviewer for their comment and have addressed this with the following text and the results section now reads “Furthermore, when we investigated the phosphorylation status of the C1 and C2 fragments we observed that they were not phosphorylated (Fig 3C-D) indicating that these fragments are potentially catalytically inactive, as Y789 phosphorylation has been shown to be required for phosphatase activity against SFKs.”

And discussion section with the following text

“The absence of phosphorylation at Tyr789 in these cleaved fragments suggests they may be catalytically inactive against SFK substrates. The phosphorylation of Tyr789 provides the binding site for the SH2 domain of Src during Src activation. Indeed, mutation or dephosphorylation of this residue blocks RPTPα ability to dephosphorylate Y527 in Src both in vitro and in vivo.”

In addition, the authors provided the same image in Fig. 3A and 3D. This becomes obvious when looking at the complete blot in the supplemental figure.

We agree with reviewer #1 and Fig 3D was accidently used also for Figure 3A. We have corrected Figure 3A (and supplemental figure) with the correct image.

In Fig. 4A, dimerization of the mutant phosphatase is tested. As a control, a delta-wedge mutant is added. This protein is hardly expressed which should at least be discussed.

We have addressed this with the following text and the section now reads “The detection of RPTPαΔwedge by immunoblot was consistently lower than the full-length proteins in our experiments (data not shown) but nonetheless demonstrated the importance of the wedge region but not R223 in RPTPα homodimerisation.”

Finally, in Fig. 4B the effect of the mutant on Src activation is investigated. Phosphorylation of Src-Y527 is more strongly present 10 min after serum stimulation than in presence of the wild type form. This is a surprising result but it is also surprising that the inhibitory Src-Y527 phosphorylation increased at all after serum treatment for both phosphatase variants. The considerably varying levels of phosphatase expression may be one explanation, however, also the level of activating phosphorylation of Akt is higher before stimulation than after 5 and 10 min. This rather points to a general experimental problem. It would have been less error prone to establish cell lines overexpressing RPTPα than performing transient expression experiments. Last, the tubulin blot in Fig. 4B, right panel is part of the total blot shown for RPTPα in the supplemental figure. How can it appear as a separate blot without additional bands as the last figure of the supplemental data?

To address this more specifically we generated stable HEK293 RPTPα WT and R223W cells, performed EGF stimulation, pERK1/2 analysis and have rewritten the entire section accordingly.

Reviewer #2: Points:

1. Src Tyr527 phosphorylation appears to be affected by expression of the R223W variant, in that serum stimulation results in a transient increase in Tyr527 phosphorylation for about 30 min in the presence of wild type RPTPalpha and only 10 min in the presence of the R223W variant (Fig.4). The authors claim that Src activity is affected differently by wild type and mutant RPTPalpha. This is not evident from the data provided. Tyr527 is an inhibitory phosphorylation site, which is phosphorylated by Csk. Activation of Src will lead to enhanced phosphorylation of the autophosphorylation site, Tyr416. Phosphorylation of Tyr416 is generally taken as a measure for Src activity. The authors should therefore reprobe their blots with Src pTyr416-specific antibodies to be able to conclude that Src activity is affected. It is noteworthy that both pTyr416 and pTyr527 are substrates of RPTPalpha.

We have changed the experimental design to use stable expressing RPTP� cells, EGF stimulation and pERK1/2 activation. 

Minor points:

1. The last sentence of the main text does not read well: “This results in a subtle, yet significant delay in activity potentially altering SFK signalling would be altered in the brains of affected patients.”

Due to the new data this sentence has been removed

2. Ref 9 appears to be the same as ref 20.

We thank the reviewer for their feedback, we have removed the duplicate reference accordingly.

---

## [Decision Letter · Decision Letter 1]

28 Jul 2021

PONE-D-21-04758R1

Does a rare mutation in PTPRA contribute to the development of Parkinson’s disease in an Australian multi-incident family?

PLOS ONE

Dear Dr. Sykes,

Thank you for submitting your manuscript to PLOS ONE. After careful consideration, we feel that it has merit but does not fully meet PLOS ONE’s publication criteria as it currently stands. Therefore, we invite you to submit a revised version of the manuscript that addresses the points raised during the review process.

There remain concerns which should be addressed if the authors plan to submit a revised manuscript.

We look forward to receiving your revised manuscript.

Kind regards,

Salvatore V Pizzo

Academic Editor

PLOS ONE

Journal Requirements:

Additional Editor Comments (if provided):

Reviewers' comments:

Reviewer's Responses to Questions

**Comments to the Author**

1. If the authors have adequately addressed your comments raised in a previous round of review and you feel that this manuscript is now acceptable for publication, you may indicate that here to bypass the “Comments to the Author” section, enter your conflict of interest statement in the “Confidential to Editor” section, and submit your "Accept" recommendation.

Reviewer #1: (No Response)

Reviewer #2: (No Response)

2. Is the manuscript technically sound, and do the data support the conclusions?

Reviewer #1: Partly

Reviewer #2: No

3. Has the statistical analysis been performed appropriately and rigorously? 

Reviewer #1: N/A

Reviewer #2: N/A

4. Have the authors made all data underlying the findings in their manuscript fully available?

Reviewer #1: Yes

Reviewer #2: Yes

5. Is the manuscript presented in an intelligible fashion and written in standard English?

Reviewer #1: Yes

Reviewer #2: Yes

6. Review Comments to the Author

Reviewer #1: The revised version of the manuscript does provide only limited improvements for the understanding of the R223W mutation in PTPa.

- A relevance for PD is purely speculative.

- The aberrant molecular weight of PTPa has not been explained and still is not pointed out in the manuscript. Yao et al. (ref 25) also use a carboxyterminal tag of PTPa and in Fig. 4B do show a Western blot with a PTPa with regular molecular weight (about 130 kD). In contrast to the authors’ statement Yao et al. do not show a MW for the PTPa in Fig 6E but indeed do so in their Figure S5 and thus give two different MW of the phosphatase. Nevertheless, I suggest to point out the strong change of MW of the tagged PTP in the manuscript.

- Phosphorylation and activity of PTPa fragments C1 and C2: the exposure shown in Fig. 3E is to short to allow detection of tyrosine phosphorylation (see bottom part of 3E and the relative amount of FL and C1/C2).

The authors state that “Y789 phosphorylation has been shown to be required for phosphatase activity against SFKs (ref 24,25)“. However, there is no activity assay for PTP fragments in ref 25. The paper of Zheng et al. (ref 24) has shown full activity for full length mutated Y789F-PTPa towards Raytide and MBP.

Since the changed proteolytical processing of PTPa-R223W that yields C1/C2 fragments is now the major novelty in this manuscript, it would have been appropriate to investigate the processing and the activity of the PTPa fragments.

- The following point is important but had been overlooked in the original review: The sentence in the manuscript “Interestingly, we discovered that the wedge region is indeed responsible for the stability of homodimers, as RPTPαΔwedge expressing cells were present almost completely as dimers“ is contradicting itself. If the wedge is important for dimerization, its deletion should abolish dimerization. This is in agreement with the finding of Jiang et al. (2000) that dimerization of PTPa was not possible when a similar deletion of the wedge region was made.

In addition to the low expression of the delta-wedge PTPa, the enhanced dimerization also needs to be discussed.

- Data shown in Fig. 4B: here, newly generated cell lines have been employed to get more physiological data. These data show clearly that there is no change of activity towards Src-pY527 in the PTPa mutant. The authors have a similar point of view, as they state at the end of the Results section “These data suggest that the phosphatase activity of the R233W mutant is not hampered in comparison to RPTPαWT after EGF stimulation.” It is therefore surprising to find as a head line of the last chapter of the Results section “The RPTPαR223W mutation decreases activity against SFKs but doesn’t alter homodimerisation“. Or this sentence in the Discussion “Nonetheless, the PTPRAR223W variant had altered the normal protein function.”

Taken together, the revised version leaves the reviewer with the conclusion that the PTPa-R223W mutant has no meaning for PD, and the only cell biological change for PTPa is a slightly altered proteolytical processing of the phosphatase for which the mechanism and the meaning are unknown.

Reviewer #2: In the original manuscript, the authors concluded that the Parkinson’s disease-associated mutation they identified in PTPRA in an Australian family altered proteolytic processing and activity of RPTPalpha. Using stable lines in HEK293 cells, they now report that SRC Tyr527 phosphorylation and ERK/MAPK phosphorylation were not affected. Proteolytic cleavage of the mutant was affected, but the functional consequences are not evident. Combined with incomplete penetrance of the mutation, the conclusion that this mutation represents a SNP that does not have functional implications would be equally valid.

SRC family kinases are the most prominent substrates of RPTPalpha. As indicated in my original review, the authors should assess Src Tyr416 phosphorylation as a read-out for SRC activity and hence for signaling downstream of RPTPalpha. If Tyr416 phosphorylation is not altered in response to the mutation, there is little or no news value in this report.

7. PLOS authors have the option to publish the peer review history of their article (what does this mean?). If published, this will include your full peer review and any attached files.

Reviewer #1: No

Reviewer #2: No

---

## [Author Response · Author response to Decision Letter 1]

13 Jan 2022

Reviewer #1: The revised version of the manuscript does provide only limited improvements for the understanding of the R223W mutation in PTPa.

- A relevance for PD is purely speculative.

We are not claiming PTPRAR223W is a causative factor in the development of PD, merely that it was found in a patient with PD and the mutation has an effect on the protein. The SNV results in aberrant Src signalling which is addressed in the final paragraph “Nonetheless, this putative PD associated mutation, PTPRA p.R223W does generate a functional protein which is cleaved by an unknown protease to a greater extent than the wild-type protein and results in impaired Src activation. Whether this contributes to increased risk of PD alone, or in concert with the other SNVs, still remains to be determined.”

- The aberrant molecular weight of PTPa has not been explained and still is not pointed out in the manuscript. Yao et al. (ref 25) also use a carboxyterminal tag of PTPa and in Fig. 4B do show a Western blot with a PTPa with regular molecular weight (about 130 kD). In contrast to the authors’ statement Yao et al. do not show a MW for the PTPa in Fig 6E but indeed do so in their Figure S5 and thus give two different MW of the phosphatase. Nevertheless, I suggest to point out the strong change of MW of the tagged PTP in the manuscript.

We thank the reviewer for their thorough analysis of the literature. We agree that there is discrepancy in the literature for the observed MW of RPRPα and have added the following to the text. 

“While the observed MW of RPTPα-V5/Flag is approximately 30 kDa larger than what is most commonly reported in the literature21–24 it is not uncommon for the protein to be reported larger than 130 kDa25–29 regardless of number or position of tags. ”

- Phosphorylation and activity of PTPa fragments C1 and C2: the exposure shown in Fig. 3E is to short to allow detection of tyrosine phosphorylation (see bottom part of 3E and the relative amount of FL and C1/C2).

To address this, we have included a higher exposure image to Fig 3E. There is no tyrosine phosphorylation in the region of the C1 and C2 bands. 

The authors state that “Y789 phosphorylation has been shown to be required for phosphatase activity against SFKs (ref 24,25)“. However, there is no activity assay for PTP fragments in ref 25. The paper of Zheng et al. (ref 24) has shown full activity for full length mutated Y789F-PTPa towards Raytide and MBP.

The authors again thank the reviewer for their thorough knowledge of the literature. Relevance of the Y789 to Src activity appears to still be debated in the literature, therefore we have added the following to the discussion:

“According to the tyrosine displacement model presented by Zheng and collegues38, pY789 of RPTPα is required to displace the Src SH2 domain, allowing access to pY527. However, several studies dispute this model, showing that Y789 phosphorylation is not necessary for pY527 dephosphorylation29,41–43. Meanwhile it has reported that activity of the Y789F mutant depended on substrate, with a pool of mutants still active against EGFR/Src, and PTPRA-potentiated ERK activation unaffected by Y789F mutants32. As no consensus has yet been reached as to the necessity of pY789, further work is required to verify the activity of C1 and C2.”

Since the changed proteolytical processing of PTPa-R223W that yields C1/C2 fragments is now the major novelty in this manuscript, it would have been appropriate to investigate the processing and the activity of the PTPa fragments.

The authors agree that investigation of the activity of the fragments would be appropriate, unfortunately it is outside the scope of this project. In addition, we now present new data showing hampered Src activation in the RPTPα R223W and R232W, and therefore a second point of interest for this SNV.

- The following point is important but had been overlooked in the original review: The sentence in the manuscript “Interestingly, we discovered that the wedge region is indeed responsible for the stability of homodimers, as RPTPαΔwedge expressing cells were present almost completely as dimers“ is contradicting itself. If the wedge is important for dimerization, its deletion should abolish dimerization. This is in agreement with the finding of Jiang et al. (2000) that dimerization of PTPa was not possible when a similar deletion of the wedge region was made.

In addition to the low expression of the delta-wedge PTPa, the enhanced dimerization also needs to be discussed.

We agree that our findings were in contradiction to the literature and we did not expect the results we obtained. However, we added the data because we thought it would help illustrate the assay. As the RPTPαΔwedge experiments do not add anything beneficial to the paper we have removed them from the manuscript and the figures.

- Data shown in Fig. 4B: here, newly generated cell lines have been employed to get more physiological data. These data show clearly that there is no change of activity towards Src-pY527 in the PTPa mutant. The authors have a similar point of view, as they state at the end of the Results section “These data suggest that the phosphatase activity of the R233W mutant is not hampered in comparison to RPTPαWT after EGF stimulation.” It is therefore surprising to find as a head line of the last chapter of the Results section “The RPTPαR223W mutation decreases activity against SFKs but doesn’t alter homodimerisation“. Or this sentence in the Discussion “Nonetheless, the PTPRAR223W variant had altered the normal protein function.”

The authors agree that consistency in our conclusions regarding protein function were not consistent in the manuscript and have now been corrected.

Taken together, the revised version leaves the reviewer with the conclusion that the PTPa-R223W mutant has no meaning for PD, and the only cell biological change for PTPa is a slightly altered proteolytical processing of the phosphatase for which the mechanism and the meaning are unknown.

With the new R223W and R232W data we think that this is addressed.

Reviewer #2: In the original manuscript, the authors concluded that the Parkinson’s disease-associated mutation they identified in PTPRA in an Australian family altered proteolytic processing and activity of RPTPalpha. Using stable lines in HEK293 cells, they now report that SRC Tyr527 phosphorylation and ERK/MAPK phosphorylation were not affected. Proteolytic cleavage of the mutant was affected, but the functional consequences are not evident. Combined with incomplete penetrance of the mutation, the conclusion that this mutation represents a SNP that does not have functional implications would be equally valid.

SRC family kinases are the most prominent substrates of RPTPalpha. As indicated in my original review, the authors should assess Src Tyr416 phosphorylation as a read-out for SRC activity and hence for signaling downstream of RPTPalpha. If Tyr416 phosphorylation is not altered in response to the mutation, there is little or no news value in this report.

The authors thank the reviewer for their insightful commentary. We had previously encountered issues with successful resolution of pSrcY416 by western blot. Similar problems were resolved in a separate project with a change in the lysis procedure. We have now performed pSrcY416 immunoblots on RPTPαR223W and discovered attenuated Src activity. We have also generated RPTPa-Sv4R232W (the longer isoform) stable cells which also have attenuated Src activity. These new data suggest that the patient SNV does indeed impact RPTPα and Src signalling.

---

## [Decision Letter · Decision Letter 2]

31 Jan 2022

PONE-D-21-04758R2Does a rare mutation in PTPRA contribute to the development of Parkinson’s disease in an Australian multi-incident family?PLOS ONE

Dear Dr. Sykes,

Thank you for submitting your manuscript to PLOS ONE. After careful consideration, we feel that it has merit but does not fully meet PLOS ONE’s publication criteria as it currently stands. Therefore, we invite you to submit a revised version of the manuscript that addresses the points raised during the review process.

As you can see Reviewer one has significant issues with respect to the manuscript. I would appreciate your reply to these criticisms.

We look forward to receiving your revised manuscript.

Kind regards,

Salvatore V Pizzo

Academic Editor

PLOS ONE

Reviewers' comments:

Reviewer's Responses to Questions

**Comments to the Author**

1. If the authors have adequately addressed your comments raised in a previous round of review and you feel that this manuscript is now acceptable for publication, you may indicate that here to bypass the “Comments to the Author” section, enter your conflict of interest statement in the “Confidential to Editor” section, and submit your "Accept" recommendation.

Reviewer #1: (No Response)

Reviewer #2: (No Response)

2. Is the manuscript technically sound, and do the data support the conclusions?

Reviewer #1: No

Reviewer #2: (No Response)

3. Has the statistical analysis been performed appropriately and rigorously? 

Reviewer #1: No

Reviewer #2: (No Response)

4. Have the authors made all data underlying the findings in their manuscript fully available?

Reviewer #1: No

Reviewer #2: (No Response)

5. Is the manuscript presented in an intelligible fashion and written in standard English?

Reviewer #1: Yes

Reviewer #2: (No Response)

6. Review Comments to the Author

Reviewer #1: The current revision does not provide relevant new insights or improve the manuscript essentially.

A misleading statement is found in the rebuttal when the authors state “We are not claiming PTPRAR223W is a causative factor in the development of PD”. Why then do they mention PD 13x in Abstract and Introduction alone, if not for nudging the reader to see a role for PTPalpha in the development of PD?

- Fig. 3E: it is surprising that in 3E the longer exposure generates considerable general darkening of the blot whereas the blot in 3C gets lighter ?

- Fig. 5: The analysis of the densitometric scanning is statistically not significant. Therefore, the statements about “percentages of” are meaningless. When looking at Fig. 5B and seeing the deviation of the mean, it just brings out a laughter when the authors state in the Results section values of 80.6±0.09% and 54.3±0.21%.

As Fig. 5 has been included to overcome the deficit in analysis of PTP cleavage (statement of the authors in the rebuttal) this also flops.

Reviewer #2: (No Response)

7. PLOS authors have the option to publish the peer review history of their article (what does this mean?). If published, this will include your full peer review and any attached files.

Reviewer #1: No

Reviewer #2: No

---

## [Author Response · Author response to Decision Letter 2]

6 Jun 2022

Reviewer #1: The current revision does not provide relevant new insights or improve the manuscript essentially.

A misleading statement is found in the rebuttal when the authors state “We are not claiming PTPRAR223W is a causative factor in the development of PD”. Why then do they mention PD 13x in Abstract and Introduction alone, if not for nudging the reader to see a role for PTPalpha in the development of PD?

We agree that the language in our response could have been clearer. We are not claiming that PTPRAR223W is the sole causative factor contributing to PD in this family. It likely increases risk of development, particularly in concert with the other identified mutations or environmental factors. We are not claiming that possessing the R223W mutation is a one-way ticket to PD. This is common in PD genetics for example even though SNCA and LRRK2 are robustly linked to PD, they are not 100% penetrant but are strong risk factors for developing PD (Delcambre et al., 2020; Trinh, Guella, & Farrer, 2014). This is covered by the following sentences in the discussion “While parkinsonism status of participants of gnomAD is not disclosed, it would be mistaken to assume that all PTPRA p.R223W carriers develop PD. Rather, we hypothesise that in this family, additive to their other genetic and environmental risk factors, the perturbed activity of the protein due to this variant conferred risk for developing PD before the age of 80 years.” and the last 2 sentences of the discussion “Nonetheless, this putative PD associated mutation, PTPRA p.R223W does generate a functional protein which is cleaved by an unknown protease to a greater extent than the wild-type protein and results in impaired Src activation. Whether this contributes to increased risk of PD alone, or in concert with the other SNVs, still remains to be determined.”

- Fig. 3E: it is surprising that in 3E the longer exposure generates considerable general darkening of the blot whereas the blot in 3C gets lighter ?

Figure 3C has clear and low intensity bands in the region of interest. Only moderate over exposure was required to visualise them better and the brightness/contrast was adjusted in ImageStudio after acquisition of the longer exposure. 3E has no noticeable bands and we have supplied the most over exposed image we have to illustrate this.

- Fig. 5: The analysis of the densitometric scanning is statistically not significant. Therefore, the statements about “percentages of” are meaningless. When looking at Fig. 5B and seeing the deviation of the mean, it just brings out a laughter when the authors state in the Results section values of 80.6±0.09% and 54.3±0.21%.

As Fig. 5 has been included to overcome the deficit in analysis of PTP cleavage (statement of the authors in the rebuttal) this also flops.

As the reviewer has had many issues with this data we have completely redone the experiments and to eliminate all sources of ambiguity we completed these analyses on single blots, using both rabbit anti-pSrc416 and mouse-anti-tSrc. Densitometric analysis revealed a significant reduction in pSrc416 phosphorylation in cells expressing mutant splice variant four RPTPα. A reduction in pSrc416 phosphorylation was apparent but not significant for the splice variant 1 RPTPα in the current experimental paradigm of 5 independent replicates. We have supplied all the blots from 5 independent replicates for complete transparency of these results (S1_raw_images.pdf).

Delcambre, S., Ghelfi, J., Ouzren, N., Grandmougin, L., Delbrouck, C., Seibler, P., … Grünewald, A. (2020). Mitochondrial Mechanisms of LRRK2 G2019S Penetrance. Frontiers in Neurology, 11, 881. https://doi.org/10.3389/fneur.2020.00881

Trinh, J., Guella, I., & Farrer, M. J. (2014). Disease penetrance of late-onset parkinsonism: A meta-analysis. JAMA Neurology. https://doi.org/10.1001/jamaneurol.2014.1909

---

## [Editor Report · Decision Letter 3]

5 Jul 2022

Does a rare mutation in PTPRA contribute to the development of Parkinson’s disease in an Australian multi-incident family?

PONE-D-21-04758R3

Dear Dr. Sykes,

We’re pleased to inform you that your manuscript has been judged scientifically suitable for publication and will be formally accepted for publication once it meets all outstanding technical requirements.

Kind regards,

Salvatore V Pizzo

Academic Editor

PLOS ONE
---

## [Editor Report · Acceptance letter]

18 Jul 2022

PONE-D-21-04758R3 

Does a rare mutation in PTPRA contribute to the development of Parkinson’s disease in an Australian multi-incident family? 

Dear Dr. Sykes:

I'm pleased to inform you that your manuscript has been deemed suitable for publication in PLOS ONE. Congratulations! Your manuscript is now with our production department. 

Kind regards, 

on behalf of

Dr. Salvatore V Pizzo 

Academic Editor

PLOS ONE